# A Colorimetric Sensor Enabled with Heterogeneous Nanozymes with Phosphatase-like Activity for the Residue Analysis of Methyl Parathion

**DOI:** 10.3390/foods12152980

**Published:** 2023-08-07

**Authors:** Fengnian Zhao, Mengyue Li, Li Wang, Min Wang

**Affiliations:** College of Chemistry and Materials Engineering, Beijing Technology and Business University, Beijing 100048, China; zhaofn@btbu.edu.cn (F.Z.); 2130042095@st.btbu.edu.cn (M.L.); 2230402131@st.btbu.edu.cn (L.W.)

**Keywords:** heterogeneous nanozyme, cerium oxide, gold nanoparticles, phosphatase-like activity, methyl parathion

## Abstract

In this study, a colorimetric sensor was developed for the detection of organophosphorus pesticides (OPs) using a heterogeneous nanozyme with phosphatase-like activity. Herein, this heterogeneous nanozyme (Au-pCeO_2_) was obtained by the modification of gold nanoparticles on porous cerium oxide nanorods, resulting in synergistic hydrolysis performance for OPs. Taking methyl parathion (MP) as the target pesticide, the catalytic performance and mechanism of Au-pCeO_2_ were investigated. Based on the phosphatase-like Au-pCeO_2_, a dual-mode colorimetric sensor for MP was put forward by the analysis of the hydrolysis product via a UV-visible spectrophotometer and a smartphone. Under optimum conditions, this dual-mode strategy can be used for the on-site analysis of MP with concentrations of 5 to 200 μM. Additionally, it can be applied for MP detection in pear and lettuce samples with recoveries ranging from 85.27% to 115.87% and relative standard deviations (RSDs) not exceeding 6.20%, which can provide a simple and convenient method for OP detection in agricultural products.

## 1. Introduction

Pesticides are a group of chemicals used to regulate plant growth and prevent pests, diseases, and weeds, which are the main chemical hazards to the quality and safety of agricultural products [1,2,3,4]. One of the main categories is organophosphorus pesticides (OPs), which are organic compounds containing phosphorus, mainly phosphate or thiophosphate, and are widely used in the growth of crops because of their broad spectrum and high efficiency [5,6,7]. When OPs are applied to crops, they will inevitably accumulate in the environment and in organisms. Studies show that these residues can accumulate through the food chain and inhibit the activity of cholinesterase in the body, thus causing symptoms in the central nervous system and posing a threat to human health [8,9,10]. Up until now, OPs have been reported as the main pollutants in water and plants [11,12]. Therefore, it is of great significance to detect OP residues for human safety and environmental protection.

In recent years, various methods based on gas chromatography [13], chromatography–mass spectrometry, and other confirmatory techniques have been applied for OP detection [14,15]. Although accurate and sensitive, the above methods are time-consuming and require specialist personnel to operate, which cannot meet the demand for rapid detection. As an important complement, biosensors have the advantage of being fast, field detectable, and easy to miniaturize, making rapid and on-site analysis for OPs possible [16,17,18]. Currently, various biosensors, including colorimetric [19], fluorescent immunoassay [20], and electrochemical biosensors [18,21], have demonstrated excellent analytical performance in the detection of OPs. However, most biosensors are based on the reaction between biological enzymes and OPs, such as acetylcholinesterase or organophosphorus hydrolases [22,23]. Despite their high specificity, these biological enzymes are expensive, difficult to prepare, and have poor stability [24,25,26]. Therefore, it is necessary to develop biomimetic recognition elements to broaden the practical application of biosensors.

Recently, with the development of nanotechnology, nanomaterials have opened new possibilities for wide application and marketization [27]. Due to their high stability, ease of synthesis, low cost, and high enzyme-like activity, nanozymes have a wide range of applications in environmental and food safety. In recent years, nanozymes have been used in electrochemical or optical sensors to analyze OP residues by exploiting their unique phosphatase [28] or peroxidase [29,30] activity. Although they have high sensitivity, peroxidase-like nanozymes commonly require the involvement of a substrate, such as H_2_O_2_, which may have autolysis and high toxicity [31]. Meanwhile, phosphatase-like nanozymes can hydrolyze OPs into small molecules with low toxicity directly, making catalysis simpler and more convenient. Recently, various phosphatase-like nanozymes, such as cerium oxide (CeO_2_) [32], porous hydroxy zirconium oxide (ZrOX-OH) [33], and zeolitic imidazolate frameworks (ZIFs) [34], have been explored. Among them, CeO_2_ nanozymes, including nanospheres, nanorods, and nanocubes, are inexpensive and environmentally friendly and have been regarded as prospective artificial phosphatases owing to the presence of Ce^3+^ and Ce^4+^ species on their surface [32,33]. However, the low specific surface area and weak adsorption ability may limit their catalytic performance. To address these issues, researchers have adopted the hybridization strategy to obtain higher catalytic activity [35,36,37]. Meanwhile, noble metals such as gold, platinum, and palladium are getting a lot of attention. As favorable catalysts, they can exhibit multienzyme activity, such as peroxidase-like activity and oxidase-like activity [26,29,38]. Therefore, the modification of noble metals on the CeO_2_ nanozyme may have a synergistic catalytic effect on the hydrolysis of OPs.

Inspired by this, we developed a heterogeneous nanozyme with phosphatase-like activity. Based on the hydrothermal method, the CeO_2_ nanorods (denoted as CeO_2_) were synthesized. After the calcination treatment, the porous CeO_2_ (denoted as pCeO_2_) nanozyme was obtained with higher phosphatase-like activity. In order to further improve the catalytic performance, gold nanoparticles (AuNPs) were modified on the substrate of pCeO_2_ to form the Au-pCeO_2_ nanozyme. Taking methyl parathion (MP) as the model pesticide, the catalytic performance and mechanism of Au-pCeO_2_ were studied by measuring the hydrolysis product (i.e., *p*-nitrophenol, *p*-NP). Due to the high hydrolysis performance, we constructed a Au-pCeO_2_ nanozyme-enabled colorimetric sensor. This dual-mode colorimetric sensor could be applied for the on-site detection of MP residues via a UV-visible spectrophotometer (UV-Vis) and a smartphone, which can provide a rapid and reliable approach for the detection of OP residues in agricultural products (Figure 1).

## 2. Materials and Methods

### 2.1. Reagents and Instruments

Cerium nitrate hexahydrate (Ce(NO_3_)_3_·6H_2_O) and carbendazim were obtained from Shanghai Aladdin Bio-Chem Technology Co., Ltd. (Shanghai, China). Sodium hydroxide, ethanol, and ethylene glycol (EG) were obtained from Beijing MREDA Technology Co., Ltd. Tris-HCl solution was obtained from Shanghai Sangon Biotech Co., Ltd. (Shanghai, China). MP, paraoxon, phosphoramidite, and monocrotophos were obtained from Alta Scientific Co., Ltd. (Tianjin, China). All other chemicals and reagents used were of analytical grade.

X-ray diffraction (XRD) patterns were obtained using an X’Pert Pro XRD device from XRD-6100 (SHIMADZU, Kyoto, Japan). High-resolution transmission electron microscopy (HRTEM) images were recorded on a Tecnai G2 F30 S-TWIN (FEI, Hillsboro, OR, USA). Raman spectroscopy was obtained with a laser Raman spectrometer (Horiba Jobin Yvon, Kyoto, Japan). To describe the surface electronic states and compositions, a Thermo Fisher Scientific K-Alpha+ X-ray photoelectron spectrometer (XPS) was used.

### 2.2. Synthesis of Nanozymes

A total of 1.736 g of Ce(NO_3_)_3_·6H_2_O and 19.2 g of NaOH were dissolved in 10 and 70 mL of deionized water, respectively, and then mixed and stirred in a glass vial for 30 min. After that, the mixture was transferred to an automatic temperature-controlled electric furnace and heated in a reaction kettle at 100 °C for 24 h. The obtained materials were cooled naturally to room temperature and then washed with 30 mL water and ethanol alternately by centrifugation until the pH was near neutral. After that, the prepared CeO_2_ was calcined at 400 °C for 1 h to obtain pCeO_2_ [35]. A total of 50 mg of pCeO_2_ powder was then dispersed in 25 mL of EG and sonicated for 30 min to make it homogeneous. A total of 250 μL of NaOH (1 M) in EG and 0.3 mL of chloroauric acid were added and then heated and stored in an oil bath at 140 °C for 3 h [36]. The reaction solution was cooled naturally to room temperature, and the precipitate was washed three times by centrifugation with ultrapure water. After drying in a vacuum oven at 75 °C and grinding to powder with a mortar and pestle, the Au-pCeO_2_ nanozymes were obtained and stored at room temperature.

### 2.3. Detection of MP

A total of 10 mg of Au-pCeO_2_ was first added into a 0.8 mL Tris buffer solution (pH 9.0, 10 mM). After the addition of 200 μL of MP solution with the known concentration, the mixture was reacted at 75 °C for 5 h in a metal bath. Then, the tube was centrifuged for 8 min at 13,523 rcf. The supernatant was then filtered through a 0.22 μm filter membrane and added to the centrifuge tube. Then, 200 μL of the supernatant was transferred into a 96-well plate for UV-Vis detection in a wavelength range of 300–500 nm. The absorbance was recorded at a wavelength of 400 nm. To realize the OP detection by a smartphone, the imaging of the samples was analyzed with the image analysis app “Color Picker” by reading the color information (RGB value) within each well.

### 2.4. Real Sample Analysis

Pears and lettuces were selected for the real sample analysis. They were purchased from a local supermarket in Beijing. Firstly, these samples were crushed in a blender and then weighed at 1.00 g (accurate to 0.01 g) into a 50 mL centrifuge tube. After that, the known concentration of MP was added to the samples for 0.5 h at room temperature. To extract MP from samples, 5 mL of acetonitrile and 1 g of NaCl were added and vortexed for 5 min. After that, the tube was centrifuged at 2348 rcf for 10 min. To reduce the matrix interference, 25 mg of primary secondary amine (PSA) and 25 mg of octadecyl silane (C_18_) were added to 1 mL of the supernatant and then vortexed for 3 min. After centrifugation for 10 min (2348 rcf), the obtained supernatant was filtered through a 0.22 μm filter membrane and collected as the extraction solution. The above solution was then blown dry with nitrogen and redissolved with Tris buffer (pH 9.0, 10 mM) for further analysis.

## 3. Results and Discussions

### 3.1. Comparison of CeO_2_, pCeO_2_, and Au-CeO_2_ Nanozymes

First of all, the morphology and chemical structure of three nanozymes were compared in this study. As shown in the insert of Figure 2a, the color of CeO_2_ is dark yellow, and pCeO_2_ is light yellow. After the addition of the precursor solution and the reduction process, the obtained nanozyme appears black-purple, which is due to the formation of AuNPs on the surface of pCeO_2_. From the TEM images in Appendix A, our CeO_2_ nanozyme appeared to have a rod-like structure. After the high-temperature calcination, the nanozyme had a porous rod-like morphology, indicating the successful preparation of pCeO_2_ (Appendix A). As shown in Appendix A, the length and width of pCeO_2_ were 42.91 ± 2.82 nm and 7.04 ± 0.17 nm, respectively.

After that, we analyzed the chemical structure of the nanozymes. The XRD patterns of the CeO_2_, pCeO_2_, and Au-pCeO_2_ nanozymes are shown in Figure 2a. The diffraction peaks of CeO_2_ were present at 2θ of 28.7°, 32.9°, 47.5°, and 54.5°, which is consistent with the reported literature [37]. After calcination, the peak pattern of pCeO_2_ was sharper than that of the CeO_2_ nanorods. After the modification of AuNPs, a weak diffraction peak appeared at 38.4°, indicating the successful preparation of the Au-pCeO_2_ nanozyme [39]. Moreover, our Au-pCeO_2_ nanozyme still had a similar crystal structure compared to pCeO_2_, so our nanozymes have the two features of AuNPs and pCeO_2_, which could provide the chance for the synergistic effect of heterogeneous nanozymes.

The Raman spectra were used to analyze the structural arrangement of the nanozymes. As shown in Figure 2b, these three nanozymes exhibited a prominent vibration at 455 cm^−1^, which was caused by the strong Raman-active F_2g_ vibrational mode typical of the CeO_2_ fluorite structure. Moreover, the peaks at this position were relatively similar for all three nanozymes, indicating the integrality of a rod-like structure. Due to the existence of a defect-induced (D) mode [40], CeO_2_ and pCeO_2_ showed a weak peak at 600 cm^−1^. After the loading with AuNPs, there was a significant change in the peak here, which also indicated AuNPs doping into the CeO_2_ lattice. It could be further concluded that the AuNPs were present on our prepared Au-pCeO_2_ nanozyme.

Using MP as the model substrate, the phosphatase-like catalytical performance of CeO_2_, pCeO_2_, and Au-pCeO_2_ for OPs was assessed by the yield of *p*-NP. A nanozyme with phosphatase-like activity can catalyze the generation of *p*-NP from MP. According to the *p*-NP standard curve (Appendix A), we can calculate the yield of *p*-NP. Here, three parallel experiments were set up. As shown in Figure 2c, the yields obtained for CeO_2_-, pCeO_2_-, and Au-pCeO_2_-catalyzed MP were 19.98% ± 1.25%, 27.16% ± 1.96%, and 45.65% ± 4.93%, respectively. Compared to CeO_2_, the catalytic performance of pCeO_2_ was higher, which was due to the surface defects and higher Ce^4+^ species [41,42]. After the hybridization of AuNPs, the activity of the nanozyme effectively increased, reaching levels twice as high as those of pCeO_2_. To reveal the catalytic mechanism, the distribution of Ce species on CeO_2_, pCeO_2_, and Au-pCeO_2_ was investigated by chemical valence state analysis (Appendix A). According to Appendix A and Figure 2d, the proportion of Ce^4+^ species increased after the calcination treatment for CeO_2_. This indicates that Ce^3+^ was shifted to Ce^4+^ in pCeO_2_ due to high-temperature (400 °C) calcination, which can increase the hydrolysis ability of the nanozyme [30]. In the Au 4f XPS spectra of Au-pCeO_2_, double peaks appeared and were assigned to Au 4f_7/2_ and 4f_5/2_. The peaks at 83.46 and 87.13 eV could be assigned to Au (Appendix A). Although the modification of AuNPs can lead to a decrease in Ce^3+^/Ce^4+^, the higher catalytic performance of Au-pCeO_2_ for MP may be attributed to the synergistic catalytic ability of two heterogeneous catalysts.

### 3.2. Optimization and Characterization of Au-CeO_2_ Nanozymes

Based on the above, we designed a series of Au-CeO_2_ nanozymes with different proportions of AuNPs by modulating the addition amount of the precursor solution (50 mM HAuCl_4_). In three parallel sets of experiments (Figure 3a), with the addition of HAuCl_4_ solution, the yield of *p*-NP of the obtained nanozyme increased. However, the catalytic performance of the obtained Au-CeO_2_ nanozyme became lower as the volume continued to increase. We inferred that was because more AuNPs could aggregate on the surface of pCeO_2_, which could occupy the active sites of the nanozymes. When the addition volume of HAuCl_4_ was 0.3 mL, the obtained Au-pCeO_2_ nanozyme showed higher and more stable catalytic performance with a 44.24% ± 2.62% yield of *p*-NP. Thus, the optimal addition volume of HAuCl_4_ was set at 0.3 mL.

After that, the morphology of the optimal Au-pCeO_2_ nanozyme was characterized using TEM and HRTEM (Figure 3b,c). It is obvious that AuNPs with a size of 10.52 ± 0.30 nm (Appendix A) had a favorable distribution on the surface of the Au-pCeO_2_ nanozyme, and pCeO_2_ still retained a rod-like structure with a clearly visible porous structure. Subsequently, the elemental composition of the Au-CeO_2_ nanozyme was further confirmed using EDS mapping analysis. As shown in Figure 3d, Au, Ce, and O elements are clearly visible, corresponding to the composition of the Au-CeO_2_ nanozyme and indicating the successful modification of AuNPs on the surface of pCeO_2_.

### 3.3. Catalytic Conditions of Au-pCeO_2_ Nanozyme

To improve the performance of the assay, the influence of various factors on catalytic activity was studied, including reaction temperature, reaction time, buffer pH, and the amount of Au-pCeO_2_. The optimum catalytic conditions were investigated by calculating the *p*-NP yields of MP (200 μM) under different conditions. Firstly, we assessed the influence of reaction temperatures (35, 45, 55, 65, 75, and 85 °C) on the hydrolysis performance of the Au-CeO_2_ nanozyme (10 mg) with a reaction time of 1 h. As shown in Figure 4a, the reaction temperature had a significant effect on the production of *p*-NP. With the increase in temperature, the yield of *p*-NP increased. After the temperature reached 75 °C, the yield of *p*-NP increased the most. Therefore, we selected 75 °C as the optimum reaction temperature. Then, the yields of *p*-NP under different reaction time (0.5, 1, 2, 3, 4, 5, and 6 h) were compared at 75 °C. As shown in Figure 4b, a higher *p*-NP yield was obtained with a reaction time of 5 h. Thus, the optimal reaction time was set to 5 h.

After that, we investigated the effect of pH (6, 7, 8, 9, and 10) on the activity of Au-pCeO_2_ (10 mg) under the reaction condition of 75 °C for 5 h. From Figure 4c, the absorbance at 400 nm increased when the pH value of Tris buffer tended to be alkaline. This takes place because the absorption peak of *p*-NP would shift to 310 nm under acid conditions, leading to a weak absorption peak [43]. When the pH value of the buffer solution was 9, the peak of *p*-NP was the highest. Thus, the optimum pH value of the buffer solution was 9. Finally, the amount of Au-pCeO_2_ nanozyme was studied under the above optimized conditions. As shown in Figure 4d, upon increasing the amount of Au-pCeO_2_ from 0 to 10 mg, more *p*-NP was produced by the hydrolysis action. While the yield of *p*-NP tends to decrease keeping increasing the amount of Au-pCeO_2_. We inferred that too many nanozymes had poor dispersion in the reaction system, so the optimal amount of Au-pCeO_2_ was 10 mg.

### 3.4. Detection Performance for MP

Under optimal conditions, we developed a Au-pCeO_2_ nanozyme-based colorimetric sensor and applied it for MP analysis via UV-Vis and a smartphone. First of all, the UV-Vis method was used to investigate the detection performance of our sensor. When the MP concentration was in the range of 5 to 200 μM, the absorbance at 400 nm (A_400 nm_) had a linear relationship with the concentration of MP (Figure 5a). As shown in Figure 5b, the standard curve was fitted with the MP concentration as the horizontal coordinate and A_400_ as the vertical coordinate. The linear equation was y = 0.0029x + 0.0821, with a high correlation coefficient (*R*^2^ = 0.9984), where y represented A_400_ and x represented the MP concentration. On the basis of the equation of detection limit (LOD) = 3 Sb/m (Sb is the standard deviation of UV-Vis in the blank experiment and m is the slope of the calibration curve) [33], the LOD of this method is 0.5 μM, which exhibits comparable analytical performance compared to the reported phosphatase-like nanozyme-enabled sensors (Appendix A).

In addition, the analytical performance of this sensor was also evaluated on a smartphone, which can be used for the on-site analysis of MP without complex equipment or instruments. By taking a picture of the reaction solution in a 96-well plate, the R/B value was recorded by the “Color Picker” app (Figure 5c and Appendix A). As shown in Figure 5d, the G/B value was also linearly related to the concentration of MP, with a linear equation of y = 0.0025x + 1.0515 (where y is the G/B value and x is the MP concentration) and a correlation coefficient of *R*^2^ = 0.9981. Based on these two analytical methods, the dual-model strategy proposed in this study can provide an accurate and reliable method for the on-site detection of MP.

### 3.5. Selectivity, Interference, Stability, and Recoverability

To investigate the selectivity of the prepared sensor, we compared the hydrolysis ability of Au-pCeO_2_ for methyl paraoxon, carbendazim, phosphamidon, and monocrotophos with the same concentration of MP (100 μM). As shown in Figure 6a, our nanozyme exhibited high *p*-NP yields for MP (24.16% ± 4.06%) and methyl paraoxon (65.46% ± 4.07%), while other pesticides produced almost no *p*-NP, indicating its excellent selectivity for these two OPs. The higher yield of *p*-NP for catalytic paraoxon may be due to its P=O bond, resulting in its higher toxicity, which is more easily hydrolyzed by the phosphatase [44].

In addition, the anti-interference study of our sensor was evaluated by testing the hydrolysis product (i.e., *p*-NP) of MP (100 μM) along with different coexisting substances (K^+^, Na^+^, and Ca^2+^ at 20 mM and ascorbic acid, glucose, and glycine at 10 mM). As shown in Figure 6b, the interference was negligible for most of the co-existing substances, except for ascorbic acid. The addition of ascorbic acid may change the original pH of the buffer, which is significantly different from the optimized conditions, thus affecting the yield of *p*-NP. It is worth noting that the amount of ascorbic acid in most agricultural products is less than 10 mM. Hence, our sensor has good anti-interference performance, making real sample analysis possible.

After that, the storage stability of our Au-pCeO_2_ nanozyme was investigated by testing the yield of *p*-NP for MP (100 μM) in five days at room temperature. As can be seen from Figure 6c, Au-pCeO_2_ retained 89% of the activity after five days compared to the original nanozyme, which demonstrates the satisfactory storage stability of the synthesized Au-pCeO_2_.

Recoverability is also an important indicator for catalysts in practical applications. For the recoverability study, our nanozyme was used for the hydrolysis of MP (100 μM) under optimal conditions, and then the used Au-pCeO_2_ was washed three times and dried. The recyclable Au-pCeO_2_ was then sonicated and dispersed into the buffer for the next five rounds of reaction. As shown in Figure 6d, the yield of *p*-NP was not less than 17.3% ± 2.08%, indicating our Au-pCeO_2_ nanozyme can be reused many times, offering economic benefits and convenience.

### 3.6. Real Sample Analysis

In this study, pears and lettuces were selected as the real samples. Before the analysis, no MP was detected in these samples, allowing them to be further used as blank samples for the recovery experiment. Herein, three spiking levels (25, 50, and 100 μM) were set in the blank samples. The sample pretreatment method was performed in Section 2.4. According to the UV-Vis analysis, the recoveries in the pear samples ranged from 87.47% to 104.83%, with relative standard deviations (RSDs) ranging from 1.69% to 2.69% (*n* = 3), while the recoveries in the lettuce samples ranged from 93.33% to 103.12%, with RSDs of 1.21% to 2.15% (*n* = 3) (Table 1). To verify the availability of our dual-mode strategy, the recovery experiment was also assessed by the RGB analysis, which also exhibited similar recoveries ranging from 85.27% to 115.87%, with RSDs of 1.96% to 6.20% for these two samples, demonstrating the favorable availability of our dual-mode strategy for MP detection in agricultural products.

## 4. Conclusions

In this study, a heterogeneous nanozyme with phosphatase-like activity was developed through the modification of AuNPs on the surface of a pCeO_2_ structure. Owing to the synergistic catalytic mechanism of these two kinds of nanostructures, our Au-pCeO_2_ nanozyme appeared to have favorable phosphatase-like activity, which can be used as a biomimetic hydrolase for the degradation of OPs. Based on this nanozyme, we proposed a dual-mode colorimetric sensor for the on-site detection of MP via UV-Vis and a smartphone. Under optimum conditions, this sensor showed a favorable linear relationship with the concentration of MP ranging from 5 μM to 200 µM. In addition, this dual-mode analytical strategy was successfully applied for MP detection in pear and lettuce samples with satisfying recoveries and low RSDs. This dual-mode assay has excellent recoverability, high stability, and good selectivity and does not need complicated equipment or professionals, which can provide a reliable and simple analytical method for OP residue determination in agricultural products.

## Figures and Tables

**Figure 1 foods-12-02980-f001:**
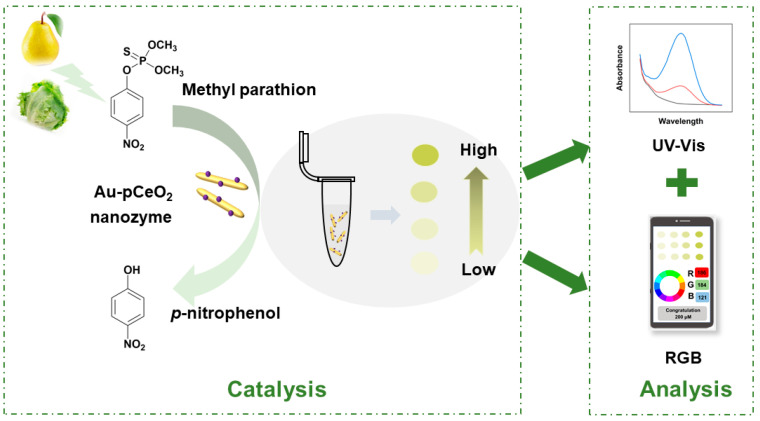
Schematic illustration of the dual-mode colorimetric sensor for MP detection based on Au-pCeO_2_ nanozyme in agricultural products.

**Figure 2 foods-12-02980-f002:**
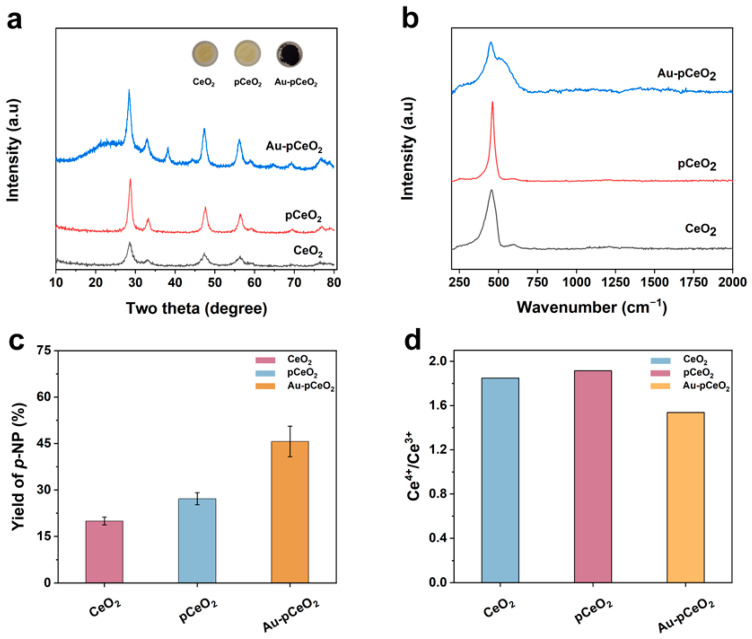
(**a**) XRD patterns of CeO_2_, pCeO_2_, and Au-pCeO_2_; the insert shows the digital photos of the prepared nanozymes. (**b**) Raman spectrogram of CeO_2_, pCeO_2_, and Au-pCeO_2_. (**c**) Yield of *p*-NP of CeO_2_, pCeO_2_, and Au-pCeO_2_ for MP (200 mM). (**d**) Proportions of Ce^3+^ and Ce^4+^ species simulated from XPS data.

**Figure 3 foods-12-02980-f003:**
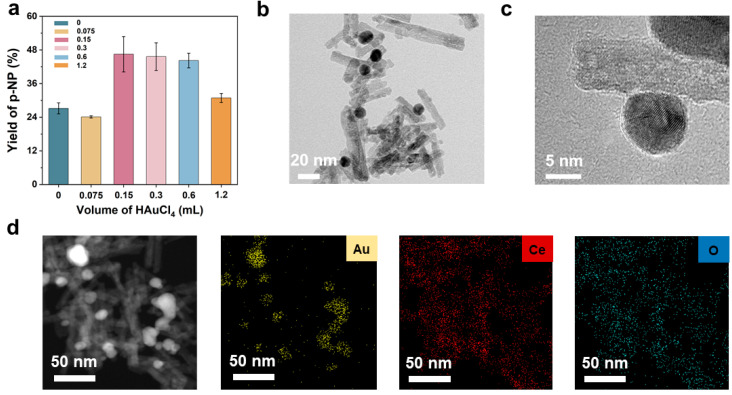
Optimization and characterization of Au-pCeO_2_ nanozyme. (**a**) Various *p*-NP yields of nanozymes at different HAuCl_4_ addition levels. (**b**) TEM and (**c**) HRTEM images of the optimal Au-pCeO_2_ nanozyme. (**d**) EDS spectra of the Au-pCeO_2_ nanozyme.

**Figure 4 foods-12-02980-f004:**
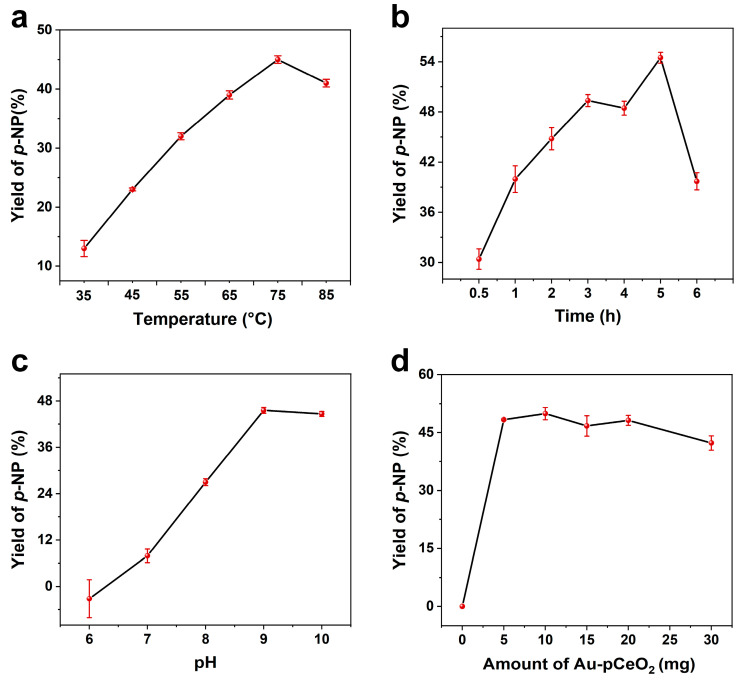
Optimization of reaction conditions of Au-pCeO_2_ nanozyme for the hydrolysis of 200 μM MP. (**a**) Reaction temperature, (**b**) reaction time, (**c**) pH value of Tris buffer, and (**d**) amount of Au-pCeO_2_ for the catalytic performance.

**Figure 5 foods-12-02980-f005:**
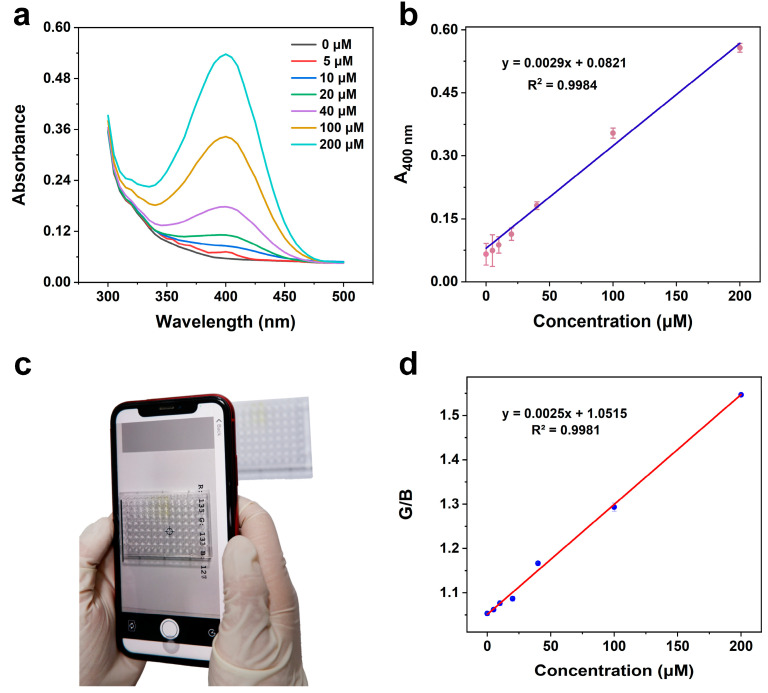
Au-pCeO_2_ nanozyme-based colorimetric sensor for MP detection. (**a**) UV-Vis spectra of various concentrations of MP (0–200 μM). (**b**) The corresponding calibration curve at 400 nm with UV-Vis spectra. (**c**) Photograph of the smartphone analysis of various concentrations of MP (0–200 μM). (**d**) The corresponding calibration curve with the G/B value on a smartphone.

**Figure 6 foods-12-02980-f006:**
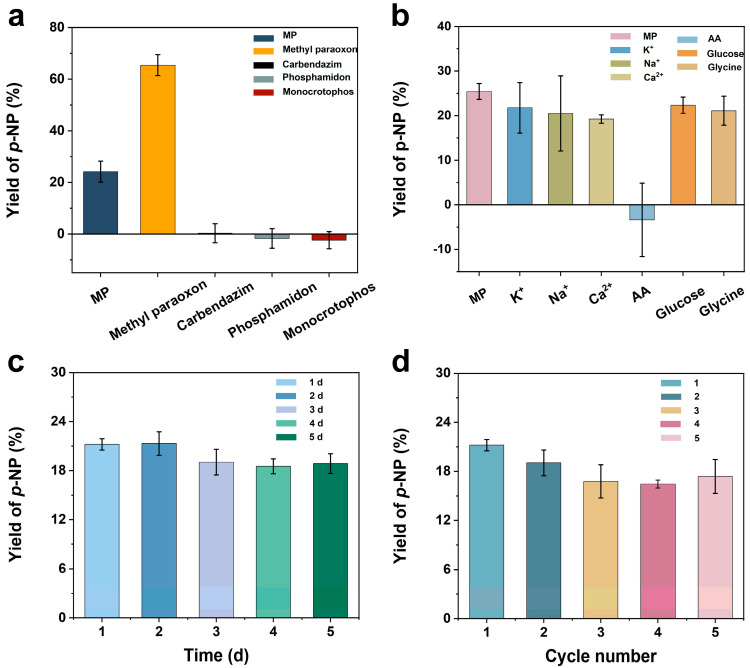
(**a**) Selectivity for various pesticides with the same concentration (100 μM); (**b**) interference resistance in the presence of coexistence substances (K^+^, Na^+^, and Ca^2+^ at 20 mM and ascorbic acid, glucose, and glycine at 10 mM); (**c**) storage stability in five days; (**d**) reproducibility study under five cycles of the proposed colorimetric sensor.

**Table 1 foods-12-02980-t001:** Recovery study of MP in pear and lettuce samples by UV-Vis and RGB analysis (*n* = 3).

Method	Samples	Added (µM)	Found (µM)	Recovery (%)	RSD (%, *n* = 3)
UV-Vis	Pear	25	26.21 ± 0.71	104.83	2.69
		50	49.22 ± 0.83	98.45	1.69
		100	87.47 ± 1.76	87.47	2.01
	Lettuce	25	25.78 ± 0.56	103.12	2.15
		50	46.67 ± 0.80	93.33	1.72
		100	95.92 ± 1.16	95.92	1.21
RGB	Pear	25	28.97 ± 0.57	115.87	1.96
		50	42.63 ± 1.60	85.27	3.76
		100	94.40 ± 4.85	94.40	5.14
	Lettuce	25	23.67 ± 0.85	94.67	3.59
		50	44.07 ± 2.73	88.13	6.20
		100	93.23 ± 2.31	93.23	2.47

## Data Availability

The data used to support the findings of this study can be made available by the corresponding author upon request.

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
