# Peer review of "A Colorimetric Sensor Enabled with Heterogeneous Nanozymes with Phosphatase-like Activity for the Residue Analysis of Methyl Parathion"

_foods, 2023, doi:10.3390/foods12152980_

Round 1
Reviewer 1 Report
The paper by Fengnian Zhao and co-authors describes the preparation of CeO2
nanorods modified with Au nanoparticles and their application for the detection of organophosphorus pesticides. In total, the paper is well written and clear, however some experimental details are missed. I think Introduction and Conclusion sections should be modified, because the developed analysis is not as simple and rapid as claimed by the Authors. Please, find my comments below.
1. Line 103 – describe washing procedure in more detail
2. Line 109 – specify centrifugation speed in rcf, as well as grinding, and drying regime
3. Line 115 – here and below use rcf instead of rpm for centrifugation speed
4. Line 115 – here and below specify regime of filtration (type of filter, pore size etc)
5. Line 128 – what is PSA and C18?
6. Line 137 – “modification of AuNPs” I don sure that Au nanoparticles were modified.
7. Line 140, Line 192 – how did you confirm formation of pores? I cannot see them in TEM images. Comparison of FigS1a and FigS1b is hardly possible because of different magnification.
8. Line 150 – “which 150 may be consistent with the catalytic results for the hydrolysis of OPs.” Please, describe relationships between peak intensity and catalytic activity in more detail.
9. Line 212 – how can you explain decrease of p-NP yield after 5 h of reaction?
10. Figure 2C – specify type of error (SD, SEM?) and number of replicates.
11. Figure 2D – strictly speaking, comparison of samples is hardly possible without information about variance. Visually difference between Ceo2 and pCeO2 is negligible.
12. Figure 3A – specify type of error (SD, SEM?) and number of replicates.
13. Figure 4C – pKa of TRIS is 7-9.2, therefore it has no buffer capacity at pH 10. Given that concentration of nanozymes was very high (10 mg/ml) decrease of p-NP yield can be explained by pH shift of solution by nanozymes themselves. Have you checked the pH of nanozymes solution at pH 10? How did you obtain TRIS buffer with pH 10? I do not sure that the pH of 10 mM TRIS is higher than 10 and can be adjusted by HCl.
14. Have you measured concentration of MP in fruits before spiking?
15. Describe how did you measured the yield of p-NP?
16. Are free gold nanoparticles (not attached to CeO2 nanorods) present in catalyst?
17. Line 308 - Analysis requires extraction, drying under N2 atmosphere, at least two centrifugation steps (one of them at a relatively high speed), and incubation for 5 h at +75 C. All these steps hardly can be performed outside specialized laboratory by non-professionals. I also cannot say that the assay is “rapid”, because only incubation step requires 5 h. Taking into account the whole procedure, analysis of single sample can take one working day. In light of this application of smartphone at the final stage of this labor-intensive procedure seems meaningless. All these steps cannot be performed without special equipment anyway. UV-Vis spectrometer is as common as centrifuges.
18. Regarding smartphone application. In reality obtaining high quality photo of 96-well plate is tricky because of non-equal angle for different wells, non-homogeneous illumination and so on. I do not sure that this method is reliable without additional equipment, which provide reproducible light conditions and fixation of the plate.
Author Response
Reviewer 1.
The paper by Fengnian Zhao and co-authors describes the preparation of CeO2 nanorods modified with Au nanoparticles and their application for the detection of organophosphorus pesticides. In total, the paper is well written and clear, however some experimental details are missed. I think Introduction and Conclusion sections should be modified, because the developed analysis is not as simple and rapid as claimed by the Authors. Please, find my comments below.
Response: Thank you so much for your positive comments. We have tried our best to revise our manuscript carefully and hope that the correction will meet with your approval.
Question 1. Line 103 – describe washing procedure in more detail.
Response: Thank you very much for your advice. We have revised the procedure to make it clear according to your advice.
Question 2. Line 109 – specify centrifugation speed in rcf, as well as grinding, and drying regime.
Response: Thank you very much for your advice. We have revised this part according to your suggestion.
Question 3. Line 115 – here and below use rcf instead of rpm for centrifugation speed.
Response: Thank you very much for your advice. We have revised the unit for centrifugation according to your advice.
Question 4. Line 115 – here and below specify regime of filtration (type of filter, pore size etc).
Response: Thank you very much for your advice. We have added detailed information on filtration in the revised manuscript.
Question 5. Line 128 – what is PSA and C18?
Response: Thank you very much for your advice. The full names of the two cleaning agents (i.e., PSA and C18) are primary secondary amine and octadecyl silane, respectively. The corresponding full names have been added to the revised manuscript.
Question 6. Line 137 – “modification of AuNPs” I don sure that Au nanoparticles were modified.
Response: We feel sorry to make you confused. The improper description has been changed in the revised manuscript.
Question 7. Line 140, Line 192 – how did you confirm formation of pores? I cannot see them in TEM images. Comparison of FigS1a and FigS1b is hardly possible because of different magnification.
Response: Thanks for your careful checks. During the synthesis process, we mainly referenced this article (DOI: 10.1021/acssuschemeng.8b02613). Compared with the reference, our pCeO2 appears a similar structure with tiny pores. We have made changes to Figure S1 and contrasted CeO2 and pCeO2. To make it clear, we also added the TEM image of pCeO2 under higher magnification and marked the pore structure with circles.
Question 8. Line 150 – “which 150 may be consistent with the catalytic results for the hydrolysis of OPs.” Please, describe relationships between peak intensity and catalytic activity in more detail.
Response: Thanks for your careful checks. After careful checking, we have revised our presentation in this paper.
Question 9. Line 212 – How can you explain decrease of p-NP yield after 5 h of reaction?
Response: Thank you very much for your advice. In our study, we tested the p-NP yields under various reaction time. Results show that the p-NP yield gets the highest after 5 h of reaction. When the reaction time increases, the yield tends to be decreased. Regarding this interesting experimental result, we speculate that it may be due to the excessive p-NP generated over too long time blocking the reaction site of the nanozyme (DOI: 10.1016/s1872-2067(20)63552-5), and the hydrolysis product may be degraded during the long-time treatment (DOI: 10.1021/acs.oprd.2c00403). In the future, we will take this point as one of our research directions.
Question 10. Figure 2C – specify type of error (SD, SEM?) and number of replicates.
Response: Thank you very much for your advice. During the experiments, we all verified the accuracy of the results through three sets of parallel experiments. In the revised manuscript, we added the corresponding description.
Question 11. Figure 2D – strictly speaking, comparison of samples is hardly possible without information about variance. Visually difference between CeO2 and pCeO2 is negligible.
Response: Thank you very much for your advice. In this study, we first characterized the synthesized substances by XPS and then split the peaks to get the ratio of elements to study the relationship between Ce valence changes and phosphatase-like activity. we refer to these two articles to measure the Ce valence state by XPS analysis (DOI: 10.1021/acssuschemeng.8b02613; DOI: 10.1002/cnma.202000132). Even though the changes in the Ce valence state are relatively small, the changes in the ratio of Ce4+/Ce3+ are obvious, which plays an important part in the nanozyme activity. Different from the measurement of p-NP yield, the parallel experiment is commonly not involved in XPS analysis. We totally understand the advice. In our next work, we will design the experiment strictly.
Question 12. Figure 3A – specify type of error (SD, SEM?) and number of replicates.
Response: Thank you very much for your advice. We have revised the manuscript to make it clear according to your advice. During the experiments, we all verified the accuracy of the results through three sets of parallel experiments. In the revised manuscript, we added the description of yield and SD.
Question 13. Figure 4C – pKa of TRIS is 7-9.2, therefore it has no buffer capacity at pH 10. Given that concentration of nanozymes was very high (10 mg/ml) decrease of p-NP yield can be explained by pH shift of solution by nanozymes themselves. Have you checked the pH of nanozymes solution at pH 10? How did you obtain TRIS buffer with pH 10? I do not sure that the pH of 10 mM TRIS is higher than 10 and can be adjusted by HCl.
Response: We sincerely thank you for your careful reading. In this paper, the 1 M Tris-HCl solution (pH=10) was purchased from the reagent company directly and then diluted with deionized water for the reaction. In our experiment, the pH of 10 mM Tris-HCl diluted only with water was around 9. To get a higher pH, the NaOH solution was added to adjust the pH value. In our study, the addition of Na+ could cause negligible interference. To eliminate the ambiguity, the buffer solution was renamed Tris buffer instead of Tris-HCl solution in the revised manuscript.
Question 14. Have you measured concentration of MP in fruits before spiking?
Response: Thank you very much for your question. We have measured the sample to ensure that it does not contain MP. Therefore, the purchased samples can be used as blank samples directly in the recovery experiment.
Question 15. Describe how did you measured the yield of p-NP?
Response: We sincerely thank you for reminding us of this important point. In this study, the standard curve of the change of p-NP concentration relative to absorbance was obtained (Figure S3). Our nanozyme with phosphatase-like activity can catalyze the generation of p-NP from MP, which has a corresponding OD value at 400 nm. The concentration of p-NP can be calculated according to the standard curve, and then the yield of p-NP can be calculated according to the original concentration of MP.
Question 16. Are free gold nanoparticles (not attached to CeO2 nanorods) present in catalyst?
Response: We sincerely thank you for your careful reading. Owing to the different densities, a vast number of free gold nanoparticles could be washed during the centrifugation, which can make sure the formation of our hybrid nanozymes (gold nanoparticles attached to CeO2 nanorods).
Question 17. Line 308 - Analysis requires extraction, drying under N2 atmosphere, at least two centrifugation steps (one of them at a relatively high speed), and incubation for 5 h at +75 C. All these steps hardly can be performed outside specialized laboratory by non-professionals. I also cannot say that the assay is “rapid”, because only incubation step requires 5 h. Taking into account the whole procedure, analysis of single sample can take one working day. In light of this application of smartphone at the final stage of this labor-intensive procedure seems meaningless. All these steps cannot be performed without special equipment anyway. UV-Vis spectrometer is as common as centrifuges.
Response: Thank you very much for your advice. In order to explore the better catalytic performance of nanozyme, we set a longer time and more appropriate temperature. In fact, our nanozyme can hydrolyze the MP in a short time. In addition, we mainly focus on the detection process. Compared with the traditional analytical technique, such as LC-MS/MS, our detection method can be used for rapid and on-site detection via a more portable device. In the future, we are devoted to investigating a more simple and rapid analytical method for pesticide residue.
Question 18. Regarding smartphone application. In reality obtaining high quality photo of 96-well plate is tricky because of non-equal angle for different wells, non-homogeneous illumination and so on. I do not sure that this method is reliable without additional equipment, which provide reproducible light conditions and fixation of the plate.
Response: Thank you very much for your advice. At present, some literature has reported the smartphone-based method for target analysis (DOI: 10.1016/j.jhazmat.2022.129199; DOI: 10.1016/j.jhazmat.2023.131171). Like these references, our study mainly focuses on the possibility of portable devices, such as smartphones for on-site screening. In our study, we also compared the analytical results, such as the recovery experiment, with the UV-Vis method. Results show that our smartphone-based method has reliable performance.
Reviewer 2 Report
In this research the authors developed colorimetric sensor for the detection of organophosphorus pesticides (OPs) using a heterogeneous nanozyme with phosphatase-like activity.
Findings, which are reported in this manuscript, are interesting and well-illustrated by results. The research is interesting from technological point of view. Some minor corrections and improvements (see comments):
- The introduction should be improved by highlighting more cited works related to the topic of paper
- after each picture, you should explain in more detail what is shown in the picture
- Please please provide a more detailed explanation Figure 2d
how did you measure the ratio Ce4+/Ce3+? the change in oxidation state could be measured by an electrochemical method - cyclic voltammetry
(example https://doi.org/10.1021/acs.jpcc.0c02973)
- please measure the detection limit in the model system and in the real system. speaking of real samples, have you tried treating real samples with your chosen pesticide and then measuring it? without adding pesticide, as explained in section 3.6.
- can any smartphone be used for measurement? please indicate exactly what the performance of the smartphone used is.
Author Response
Reviewer 2.
In this research the authors developed colorimetric sensor for the detection of organophosphorus pesticides (OPs) using a heterogeneous nanozyme with phosphatase-like activity. Findings, which are reported in this manuscript, are interesting and well-illustrated by results. The research is interesting from technological point of view. Some minor corrections and improvements (see comments).
Response: Thank you so much for your positive comments. We have tried our best to revise our manuscript carefully and hope that the correction will meet with your approval.
Question 1. The introduction should be improved by highlighting more cited works related to the topic of paper.
Response: Thank you very much for your advice. We have revised this part according to your suggestion.
Question 2. After each picture, you should explain in more detail what is shown in the picture.
Response: We sincerely thank you for your careful reading. We have explained in more detail what is shown in the picture.
Question 3. Please provide a more detailed explanation Figure 2d.
Response: Thank you very much for your advice. In our study, the Ce valence state was investigated by XPS characterization and peak splitting process. The detailed data was shown in Figure S4. In order to make it clearer, we have modified the diagrammatic notes of Figure 2d.
Question 4. How did you measure the ratio Ce4+/Ce3+? the change in oxidation state could be measured by an electrochemical method - cyclic voltammetry (example https://doi.org/10.1021/acs.jpcc.0c02973)
Response: Thank you very much for your advice. In this study, we mainly refer to these two articles to measure the ratio of ratio Ce4+/Ce3+ by XPS analysis (DOI: 10.1021/acssuschemeng.8b02613; DOI: 10.1002/cnma.202000132). We first characterized the synthesized substances by XPS and then split the peaks to get the ratio of elements to study the relationship between Ce valence changes and phosphatase-like activity. Thank you for your suggestion, your suggested method can provide a new idea for our future experiments.
Question 5. Please measure the detection limit in the model system and in the real system. speaking of real samples, have you tried treating real samples with your chosen pesticide and then measuring it? without adding pesticide, as explained in section.
Response: Thank you very much for your advice. According to the formula of detection limit (LOD) = 3Sb/m (Sb is the standard deviation of UV-Vis in the blank experiment and m is the slope of the calibration curve), the LOD of our method is 0.5 μM. Before the recovery study, we tested the MP residues with the confirmed technique. Results show the MP is not detected. Therefore, our samples can be used as blank samples for the recovery study. It should be noticed that it is difficult to obtain the available positive samples from the local supermarket. In this regard, we only did the recovery study for methodological evaluation.
Question 6. Can any smartphone be used for measurement? please indicate exactly what the performance of the smartphone used is.
Response: We sincerely thank you for your careful reading. Our smartphone-based analytical method can be achieved on an unlimited-model Apple phone or even iPad with picture-taking capability, in which the “color picker” app can be successfully loaded and compatible.
Round 2
Reviewer 1 Report
I find this article suitable for publication in its present form.